# Geometric Morphometric Wing Analysis of Avian Malaria Vector, *Culiseta longiareolata*, from Two Locations in Algeria

**DOI:** 10.3390/insects13111031

**Published:** 2022-11-08

**Authors:** Mounir Boumaza, Brahim Merabti, Yasmine Adjami, Mohamed Laid Ouakid, Thaddeus M. Carvajal

**Affiliations:** 1Department of Biology, Faculty of Sciences, Badji Mokhtar University, B.P. 12, Annaba 23000, Algeria; 2Ecology Laboratory of Marine and Coastal Environments (EMMAL), Badji Mokhtar University, Annaba 23000, Algeria; 3Laboratory of Genetic, Biotechnology and Valorization of Bioresources (LGBVB), University of Biskra, Biskra 07000, Algeria; 4Department of Biology, College of Science, De La Salle University, Manila 1004, Philippines; 5Ehime University-De La Salle University International Collaborative Research Laboratory, Laguna Campus, De La Salle University, Laguna 4024, Philippines

**Keywords:** Culicidae, geometric morphometrics, Algeria, larval habitats, climate

## Abstract

**Simple Summary:**

*Culiseta longiareolata* (Macquart 1838) is a cosmopolitan mosquito species and is considered to be an important vector in the transmission of avian malaria, tularemia, and arboviruses. The present study investigates the population structure of *Cs. longiareolata* from different bioclimatic and larval habitat types using a wing geometric morphometric approach. The main findings of our study showed that these environmental factors shape the population structure of *Cs. longiareolata,* most especially in male mosquitoes. This further deepens our understanding of how vector mosquitoes such as *Cs. longiareolata* adapt and thrive in different environmental conditions.

**Abstract:**

The application of geometric morphometry on mosquito wings (Culicidae) is considered a powerful tool for evaluating correlations between the phenotype (e.g., shape) and environmental or genetic variables. However, this has not been used to study the wings of the avian malaria vector, *Culiseta longiareolata*. Therefore, the goal of this study is to investigate the intra-specific wing variations between male and female *Cs. longiareolata* populations in different types of larval habitats and climatic conditions in Algeria. A total of 256 *Cs. longiareolata* mosquito samples were collected from January 2020 to July 2021 in three cities (Annaba, El-Tarf, and Guelma) of northeastern Algeria that have two distinct climatic condition levels (sub-humid and sub-arid) and different types of larval habitats (artificial and natural). Nineteen (19) wing landmarks (LMs) were digitized and analyzed based on geometric morphometry. Our results revealed differences in the wing shape of female and male mosquito populations, indicating sexual dimorphism. Moreover, canonical variance analysis (CVA) showed that factors, such as climatic conditions and type of larval habitats, also affect the wing shape of female and male *Cs. longiareolata* mosquito populations. Furthermore, the wing shape of male populations was more distinct compared with female populations.

## 1. Introduction

The mosquito *Culiseta longiareolata* (Macquart 1838)was first reported in Algeria (North Africa) at the beginning of the 20th century [1]. It was found in all types of natural habitats (e.g.,lagoon depressions, valleys, ditches) throughout the country [2]. It is widely distributed in the south of the Palearctic and Mediterranean regions, as well as in Europe and Asia [3,4].More specifically, they have been found in diverse locations, from freshwater rocks and pools to plastic containers, casks, tire basins, and fountains [5]. They are easily distinguished from other *Culiseta* species because of their white stripes and the points on their legs, head, and thorax [6]. They aggregate in agricultural and urban areas and have been described as an ornithophilic species that rarely bite humans [7,8,9,10]. However, previous studies have reported that this mosquito species could be a possible vector of the bacteria that cause Malta fever (brucellosis) [11] and westernencephalitis virus [12].

Environmental conditions, including important climatic variables, affect the distribution and abundance of many mosquito species, including those of concern to human health [13].For example, *Aedes albopictus* mosquitoes found in urban sites were observed to have decreased larval survival, smaller body sizes, and lower per capita growth rates compared with those in rural sites [14]. On the other hand, *Culex pipiens* mosquitoes were seen to have an increased adult density and longer breeding season in areas that have higher temperatures [15]. To date, there is little information about the impact of environmental factors on the growth and phenotype of emerging *Cs. longiareolata* adults when compared with other mosquito species. Therefore, knowledge regarding the relationship between habitats, environmental factors, and mosquito plasticity is essential for developing effective mosquito control methods.

Shape analysis is an approach that allows for a better understanding of the different causes of variation and morphological transformation [16] in organisms. Studying the biological shape or phenotype of these insects will allow us to understand the ecological, developmental, and genetic alterations of insects in changing environmental conditions. Previous studies have demonstrated that insect wings provide a link between phenotype and the environment using geometric morphometry [17,18]. More specifically, studies on *Aedes albopictus* [19], *Anopheles darling* [20], and *Aedes aegypti* [21] have all demonstrated that there are variations in mosquito populations that are influenced by environmental factors, manifesting in sexual dimorphism, differences in wing shape, and variations in size.

There are limited studies on the avian malaria vector, *Cs. longiareolata*. This study is the first to report its microevolution using geometric morphometry. The primary aim of the study is to determine the intra-specific wing variation between male and female *Cs. longiareolata* mosquitoes in different climatic and larval habitat types.

## 2. Materials and Methods

### 2.1. Study Area

Algeria is in the north of Africa, with a surface of 2382 million km^2^. The collection of mosquito samples was conducted between January (2020) and July (2021) in thirteen areas distributed in three cities: Annaba (36°54′ N, 07°44′ E), El-Tarf (36°45′ N, 8°18′ E), and Guelma (36°14′ N, 07°15′ E) (Figure 1). The country has six different climatic zones: hyper-humid, humid, sub-humid, sub-arid, arid, and Saharian [22,23,24]. Appendix A shows the information regarding mosquitoes collected by locality, geographic coordinates, climate, weather data from actual sampling year, larval habitat nature, and sex.

The city of Guelma is considered sub-arid. The mosquito samples were taken in Ain Makhlouf, where temperatures reach 10.33 °C and 26.1 °C in the winter and summer, respectively [25], while the annual precipitation is 484.37 mm, and the average humidity is 63%.

The cities of Annaba and El-Tarf are considered sub-humid. Annaba has an average temperature of 18.4 °C throughout the year, The warmest month is August (30.3 °C), and the coldest month is February, with an average temperature of 14.3 °C. The average humidity is 70.41%, and its precipitation averages 712 mm (650 and 1000 mm/year) per year [26]. On the other hand, El-Tarf has a mean temperature of 12 °C and 28 °C during winter and summer, respectively, and its mean annual precipitation reaches 700 mm [27]. The average humidity is 70%. The month with the highest relative humidity is March (76%), and the month with the lowest relative humidity is July (60%).

### 2.2. Mosquitoes Sampling and Identification

For this study, 256 immature stages were sampled simultaneously from 35 collection sitesfrom January (2020) until July (2021) in two different climatic zones using a standard dipper described by Papierok et al. [28] in artificial and natural larval habitats. Artificial larval habitats included animal water dishes, bottles, and 200 L tanks, while natural larval habitats included ground pools, ditches, and swamps. The average distance between the collection sites in the two sub-humid and sub-arid bioclimatic zones is between 80 and 110 km, respectively. L4 larvae and pupae were selected and placed in a plastic pan (500 mL) with their original water containing organic material until reared to adults. No additional larval food was introduced for rearing. Species identification and sex determination were performed using the Mediterranean African Culicidae software [29] and the dichotomous key of the Culicidae of Morocco [30].

### 2.3. Wing Preparation

The rightwing of each male and female mosquito was detached from the thorax and processed following the protocol described by Lorenz and Suesdek [31], with some modifications. Each wing was bathed using 3% sodium hypochlorite (NaClO) for 20 min. The scales were removed from the wings using cotton swabs, followed by a 99.5% ethanol wash. Afterward, the wings were bathed in an acid fuchsin solution for 1 h and then washed with 70% ethanol twice. Finally, the wings were mounted between a slide and a cover slip with a drop of Euparal© mounting medium (Carl Roth, Karlsruhe, Germany). The mounted wings were photographed using a camera(SI 3000 version I8, Warpsgrove Ln, Chalgrove OX44 7XZ, UK) with a 10× magnification on a stereoscopic microscope (CETI Steddy Stereo Trinocular Microscope, Warpsgrove Ln, Chalgrove OX44 7XZ, UK). In total, 19 landmarks (LMs) [32] were identified and digitized using Tps Dig2 (V2.31) software [33] to generate the Cartesian coordinates in two dimensions for each individual mosquito (Appendix A, Appendix A).

### 2.4. Data Analysis

The measurement error was tested by comparing three sets of digitization as described by Arnqvist [34]. This test was performed on 256 individual wing images three times by the same researcher. The analysis of variance (ANOVA) and multivariate analysis of variance (MANOVA) tests were applied to compute the variation among replicate measurements and to evaluate the computed wing size and shape of *Cs. longiareolata* varied across the three landmark collections.

To determine the wing size variation, an isometric estimator, known as the centroid size, was computed from the 19 landmarks [35]. The centroid sizes between males and females based on their climatic and larval habitat factors were subjected to comparative analysis using a parametric T-test, Kruskal–Wallis test with a post hoc analysis, and a Mann–Whitney test with Bonferroni correction using the SPSS V23.0 software [36]. Centroid sizes were visualized by using a box-and-whisker diagram to indicate the significant differences among the comparisons. An allometric effect of the wing size on the wing shape was also performed for both males and females using a multivariate regression analysis of the Procrustes coordinate of wing centroid size. The statistical significance of the allometric effect was determined by non-parametric permutation testing with 10,000 randomizations.

To determine the wing shape variation, a generalized least-squares Procrustes superimposition algorithm was performed [37]. Covariance matrices were generated for each superimposed dataset to allow for the exploration of variations via the principal component analysis (PCA). The canonical variate analysis (CVA) with a permutation of 10,000 randomizations was carried out to test the pairwise Mahalanobis distances (MD) between or among the different groups of males and females based on their climate and larval habitats. Thin-plate spline and wireframe diagrams [35,38] were generated to explore the intra-specific wing shape variation influenced by climate and natural larval habitats in male and female mosquito populations. The analyses of wing size and shape were all conducted using the MorphoJ software version 1.07 [39].

## 3. Results

### 3.1. Repeatability of Landmarks and Test for Allometry

Three measurements of the size and shape of male and female *Cs. longiareolata* wings showed good precision in the digitization of the landmarks: size (Male: F = 0.04; P = 0.95; Female: F = 0.69; P = 0.59) and shape (Male: F = 0.44, Pillai’s trace = 0.33, P = 1.00; Females: F= 0.79, Pillai’s trace = 0.39, P = 0.79). This indicates that the differences found in the morphology of the wings are due to climatic and habitat factors, not from measurement error. Moreover, the results of the allometry indicate that the wing shape variance explained by size was 23.21% for female and 19% for male mosquitoes. Lastly, our analysis revealed the contribution of the centroid size to the variation in the wing shape of *Cs. longiareolata* was significant (*p* < 0.0001) for both sexes. Therefore, we removed the allometric effect from all analyses to analyze the wing size and shape separately.

### 3.2. Sexual Dimorphism

The wing centroid size in the male mosquito population varied from 3.21 mm to 4.92 mm, while the female mosquito population varied from 3.81 mm to 6.72 mm (Figure 2a). Further analysis also showed that the wing centroid sizes between male and female mosquito populations were significantly different from each other (t = 10.24; *p* < 0.0001).

Figure 2b shows the canonical variate analysis (CVA) plot where the wing shape of male and female mosquito populations was distinctly separate (MD = 7.42; *p* < 0.0001). Additionally, thin-plate splines (Figure 3a) demonstrated that the shape of the male wings was narrow and lengthened, while the female wings were wider and shorter (Figure 3b). Landmarks 2 (intersection of costa), 16 (radio-sectoral vein), and 19 (origin of radius branches 2 and 3) appeared to be the most important landmarks in the shape difference between males and females.

### 3.3. The Effect of Sub-Arid and Sub-Humid Climatic Conditions

Figure 4a shows the centroid sizes of sub-arid and sub-humid female mosquito populations. Further analysis showed that there was no significant difference in the female wing centroid sizes between the two climatic conditions. However, the CVA plot indicated that the wing shape of female mosquitoes in sub-arid and female mosquitoes in sub-humid climatic conditions was distinct (MD = 1.57; *p* < 0.0001) (Figure 4b).

The results were the same with the male populations. No significant differences were found in their wing centroid sizes between sub-arid and sub-humid climatic conditions (Figure 4c), and the CVA plot indicated a degree of distinction between the two climatic conditions (MD = 2.81; *p* < 0.0001) (Figure 4d).

### 3.4. The Effect of Natural and Artificial Larval Habitats

No significant differences were found in the wing centroid sizes between female mosquito populations from natural and artificial larval habitats (Figure 5a). However, the CVA plot revealed significant differences in the wing shape of females from natural and artificial larval habitats (MD = 1.68; *p* < 0.0001) (Figure 5b).On the other hand, there were no significant differences in the wing centroid sizes between male mosquito populations from natural and artificial larval habitats (Figure 5c). However, a significant distinction in wing shape was found in males from natural and artificial larval habitats (MD = 2.12; *p* < 0.0001) (Figure 5d).

### 3.5. Combined Effect of Larval Habitat and Climatic Conditions in the Wing Size and Shape

In females, it was revealed that there were significant differences in wing sizes depending on the factors of climatic patterns and type of larval habitats (X^2^ = 74.48, df = 3, *p* < 0.000) (Figure 6a). Post hoc analysis showed significant differences among all groups except for female populations from artificial larval habitats in the sub-arid climate (A.A) and natural larval habitats in the sub-humid climate (N.H). In males, the results also showed that there were significant differences in wing sizes (X^2^ = 32.58, df = 3, *p* < 0.000) (Figure 6b) depending on the factors of climatic patterns and type of larval habitats. Post hoc analysis showed significant differences among all groups except for male populations from artificial larval habitats in the sub-arid climate (A.A) and natural larval habitats in the sub-humid climate (N.H).

The results for wing shape showed significantly structured populations based on climatic patterns and type of larval habitats in female (F = 2.05; Pillai’s trace = 1.02; *p* < 0.0001) and male (F = 1.82; Pillai’s trace = 1.67; *p* < 0.0001) mosquitoes. Furthermore, CVA plots indicated that the wing shape of male populations based on the two factors was more distinct compared with that of females (Figure 6c,d).

## 4. Discussion

### 4.1. Sexual Dimorphism

Previous studies have demonstrated the sexual dimorphism of wing size and shape analyses in insects [40,41,42], especially in the Culicidae species [19,43,44,45,46,47,48]. However, this is the first study that demonstrated sexual dimorphism in *Cs. longiareolata* using wing size and shape. More specifically, the wing shape of male *Cs. longiareolata* was narrow and long, while female wings were wider and shorter. The highest differences between female and male wing shapes were found in the middle and distal regions of the wing (LMs2, 16, and 19) based on thin-plate splines. Our findings are consistent with the observation of other mosquito species such as *Aedes* [19,43,49,50,51], *Anopheles* [52], and *Culex* [44,53,54].

These differences could be attributed to the important role of wing shape in reproductive functions in that it increases performance in flight-based mating tactics [55]. As evidenced by scrambling damselflies (*Lestes sponsa*), males with broader and shorter forewings had better mating success [56]. Moreover, mosquito wing geometry is involved in species-specific wing beat frequencies that mediate different mating behaviors [57], as seen in the production of courtship sounds in Orthoptera species [58]. Changes in wing shape are also influenced by the environment and impact their ability to locate oviposition sites [59], which affects their reproductive capacity.

The adaptation of each sex to various reproductive functions is reflected in sexual size dimorphism [60]. A female mosquito’s big wings could be attributed to long-distance dispersal because of its host-seeking behavior. Interestingly, the average wing length of nulliparous host-seeking females was significantly smaller than the wing length of parous host-seeking females [59,61].On the other hand, the small-sized wing in males could be attributed to their short development time, resulting in smaller body size [19] and their short-distance dispersal by remaining near larval habitats in seeking mating partners [45,49,51,62]. The female body size has also been associated with its capacity as a vector for diseases [60], and it has been shown that a population of mosquitoes with a large average body size may have a higher vector capacity than a population with a small average body size.

### 4.2. Effects of the Climatic Conditions and Larval Habitats

In both sexes, our study showed that the wing shape of *Cs. longiareolata* in sub-arid populations was distinct from that in sub-humid populations. The influence of climatic conditions could be attributed to one or a combination of several factors, including temperature, humidity, and precipitation. Other environmental factors such as different microhabitats [63], environmental heterogeneity of the geographical area [50], and climatic effects (such as temperature) [64] could have also influenced these results. Studies on damselflies [65], *Drosophila melanogaster* [66], and *Drosophila willistoni* [67] have reported that temperature has a strong influence on the wing shape of these insects. Other studies on other mosquito species such as *Cx. coronator* [68], *Anopheles albimanus* [69], *An. funestus* [70], and *An. superpictus* [71] also support this notion that climate plays a role in shaping the wings of many insect populations.

In addition, our results indicated that there were no significant changes in wing sizes between the two climatic conditions for both sexes despite the differences in rainfall and mean air temperature. This may be due to the latitude of the locations where the mosquitoes originated. Generally, insects react to environmental conditions according to James’s rule (increase in body size with latitude or decrease in temperature) or Bergmann’s rule (the size decreases with latitude, as a consequence of a shorter favorable season and an increase in developmental metabolism) [19]. In our study, the sampling sites had two different bioclimatic levels but belonged to the same latitude. Unfortunately, our analysis only measured the mosquito’s wings. As a result, we were unable to verify this claim, leaving the door open for future research. However, Gómez and Márquez [68] posited that relative humidity (RH) and elevation could explain the variation in centroid sizes. For various mosquito models, the influence of RH on centroid size has been described [72,73]. Rainfall differences have also been reported to alter the centroid size of *An. colzzii* in Burkina Faso [74] and *Aedes albopictus* in central Argentina [75]. These studies all support the claim that RH (and possibly, elevation) can affect centroid size.

On top of that, larval habitats and larval competition have been reported to affect the body size of mosquitoes [71,73,76]. In this study, differences were observed in male and female wing shapes between mosquitoes found in natural and artificial larval habitats, which is consistent with *Anopheles cruzii* [77], *Ae. albopictus*, and *Ae. aegypti* [78,79]. The density, plant material, and competitive pressure in urban larval habitats can likewise affect the development of the larvae. This is supported by a study that showed that water containing more plant material accelerates Ae. albopictus larvae development [80,81].In this study, no significant differences in the *Cs. longiareolata* wing sizes were found between natural and artificial larval habitats for both sexes. Stephens and Juliano [82] suggested that the relationship between size and rearing conditions differed between the species and that size could not be used to determine larval rearing conditions. However, Oliveira-Christe and Wilke [83] suggested that the variations found in wing size may be related to the intrinsic characteristics of larval habitats, such as temperature, water physicochemical parameters, and food availability. The temperature during rearing could also likely affect the survival and development of immature larvae and adults [84].

Furthermore, our findings indicate that there were significant differences in wing sizes and shapes depending on the factors of climatic patterns and type of larval habitats for both sexes, except between male and female populations from artificial larval habitats in the sub-arid climate (A.A) and natural larval habitats in the sub-humid climate (N.H). This may be related to several ecological, morphological, and physiological factors specific to Diptera in general. Rapid deformations in wings would surely affect the rate and distance of insect dispersion and would exert pressure on the energy required for flying. This could then impact other ecological parameters, such as the success of finding mates, the kinematics of flying, dispersal, and pressure from predators [71].

### 4.3. Limitations

This study revealed phenotypic variations among the populations of *Cs. longiareolata* located in sub-humid and sub-arid regions including its larval habitat variability factors; however, there were several limitations that should be considered. First is the uneven number of mosquito samples of each sex and its corresponding environmental factors. The study tested a small number of male mosquitoes (8–42 individuals), compared with their female counterparts (18–76 individuals). The uneven sample sizes were also found in natural larval habitat mosquitoes (8–38 individuals) and artificial larval habitat mosquitoes (20–76 individuals) as well as sub-arid mosquitoes (8–35 individuals) and sub-humid mosquitoes (18–76 individuals). Therefore, the results inferred by this study could likely affect the phenotypic diversity of each population. The authors suggest that future studies should consider even sample sizes to validate our findings. Moreover, no information about factors such as larval density, water temperature, and availability of food was collected from larval habitats; therefore, we were unable to predict or estimate the magnitude of the effect of these variables on wing size. It should be noted that future studies should take into consideration these other factors as well.

## Figures and Tables

**Figure 1 insects-13-01031-f001:**
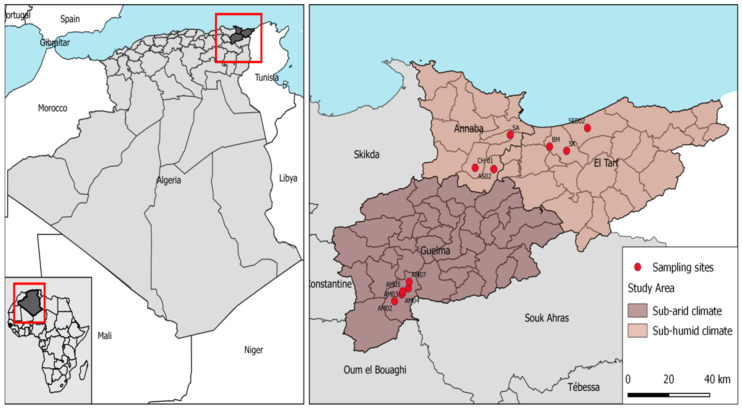
Geographic map of Algeria (**left**) and its study area as well as sampling sites indicating the sub-humid and sub-arid climate (**right**).

**Figure 2 insects-13-01031-f002:**
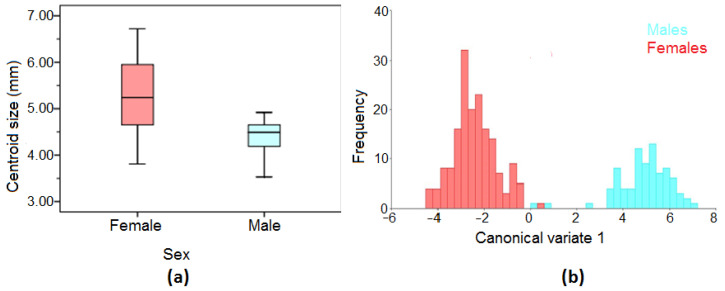
(**a**) Box diagram of centroid sizes for all females and males; (**b**) wing shape diagram of first canonical variable from the comparison of all males (blue) and females (red).

**Figure 3 insects-13-01031-f003:**
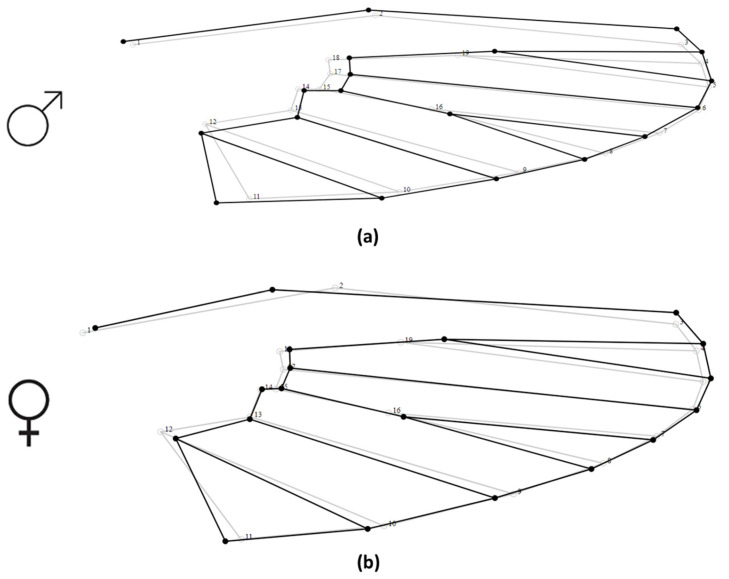
Wireframe diagram from PCA displaying the mean value of wing shape variation in (**a**) male and (**b**) female *Cs. longiareolata* mosquitoes. The light-colored wireframe represents the mean wing shape, while the dark-colored wireframe represents generated principal component analysis for the wing shapes.

**Figure 4 insects-13-01031-f004:**
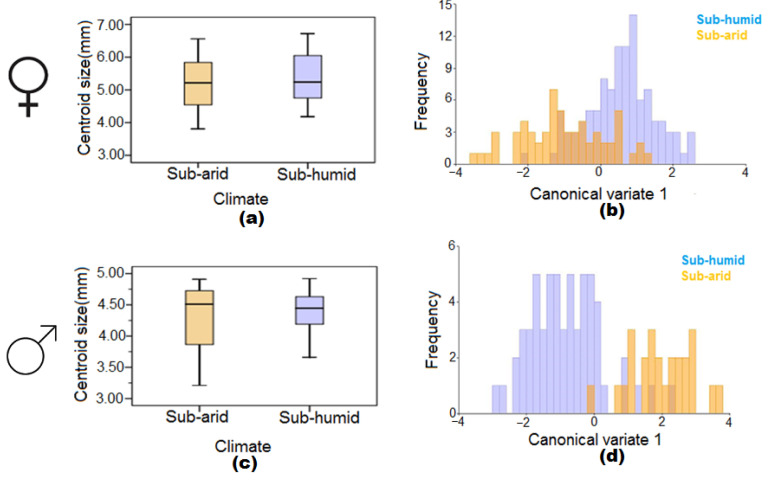
Boxplots of wing centroid size and wing shape diagram of the first canonical variable between sub-humid and sub-arid climatic conditions in female (**a**,**b**) and male (**c**,**d**) *Cs. longiareolata* mosquitoes.

**Figure 5 insects-13-01031-f005:**
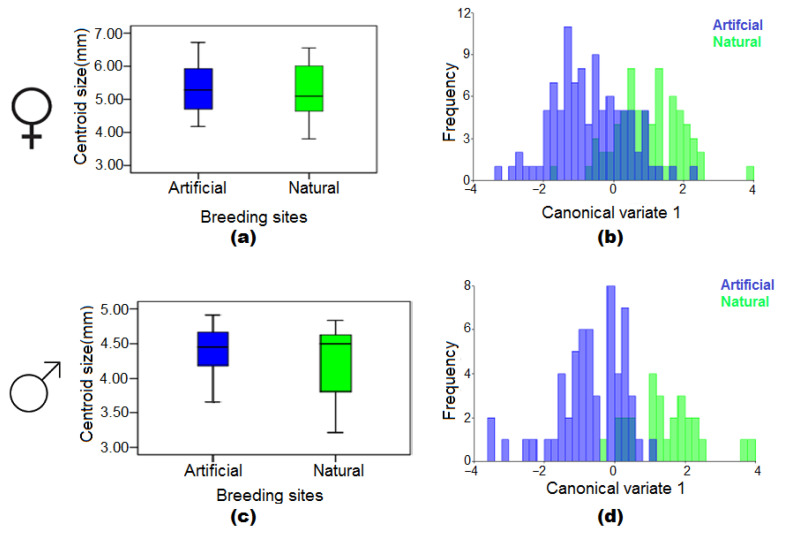
Box plots of wing centroid size and wing shape diagram of the first canonical variable between natural and artificial breeding sites in female (**a**,**b**) and male (**c**,**d**) *Cs. longiareolata* mosquitoes.

**Figure 6 insects-13-01031-f006:**
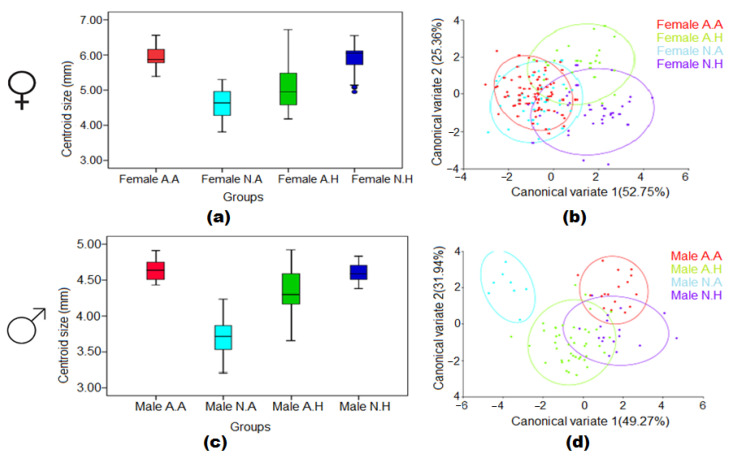
Boxplots of wing centroid size and wing shape diagram of the first two canonical variables between the interaction of climate and breeding sites in female (**a**,**b**) and male (**c**,**d**) *Cs. longiareolata* mosquitoes. Group A.A: *Artificial Breeding site*, *Climate Sub-Arid*; Group N.A: *Natural Breeding site*, *Climate Sub-Arid*; Group A.H: *Artificial Breeding site*, *Climate Sub-Humid*; Group N.H: *Natural Breeding site*, *Climate Sub-Humid*.

## Data Availability

The data presented in this study are available on request from the corresponding author. The data is not publicly available because no available repository is available to deposit the dataset.

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
