# Peer review of "Geometric Morphometric Wing Analysis of Avian Malaria Vector, Culiseta longiareolata, from Two Locations in Algeria"

_insects, 2022, doi:10.3390/insects13111031_

Round 1
Reviewer 1 Report
The manuscript uses a morphometric approach to investigate differences in specimens of Culiseta longiareolata collected in different habitats in northern Algeria. The authors investigate both shape and size variation in males and females, as well as between those form natural and artificial larval habitats and from semi-arid an semi-humid conditions. Evidence for sexual dimorphism is found, and for variation in shape between these different habitats. Plausible biological explanations are discussed.
Whilst the findings of the study are of limited importance, the approach taken is sound and the investigation will be of interest to some readers. I only have minor comments for corrections:
Line 41 – remove first and second commas
Line 44 – state that they are found in these habitats as larvae, because the following sentence describes features of the adults
Line 54 – add genus name in full
Line 59 and elsewhere – use a space between Cs. and longiareolata
Line 249 needs re-writing
Line 256 – remove "etc”
Section 3.2 and figure 3 – Can you clarify whether the PCA was performed on all specimens to look for differences between males and females? The black wireframe image in figure 3a does not appear to be the grey wireframe in 3b, so it is no clear if these images are showing the sexual dimorphism.
Paragraph starting line 279 – this section switches between shape and size, and would benefit from some revision to make it easier to follow.
Line 310 – fluctuating asymmetry is introduced here and seems out of place. Consider revising the closing sentence or add some suitable references.
Author Response
|
Code |
Comment / Suggestion |
Action taken |
|
R1-01 |
Line 41 – remove first and second commas |
modified |
|
R1-02 |
Line 54 – add genus name in full |
modified |
|
R1-03 |
Line 59 and elsewhere – use a space between Cs. and longiareolata |
modified |
|
R1-04 |
Line 256 – remove "etc” |
removed |
|
R1-05 |
Section 3.2 and figure 3 – Can you clarify whether the PCA was performed on all specimens to look for differences between males and females? The black wireframe image in figure 3a does not appear to be the grey wireframe in 3b, so it is no clear if these images are showing the sexual dimorphism. |
Thank you for this comment. First, PCA was performed for all male and female mosquito samples respectively. The grey wireframe represents the mean wing shape of male or female mosquitoes while the black wireframe represents generated principal component analysis for the wing shape of male or female mosquitoes. We included the explanation in Figure 3 (Please see lines 190 to 192).
|
|
R1-06 |
Paragraph starting line 279 – this section switches between shape and size, and would benefit from some revision to make it easier to follow. |
clarified |
|
R1-07 |
Line 310 – fluctuating asymmetry is introduced here and seems out of place. Consider revising the closing sentence or add some suitable references. |
We highly appreciate this comment. It was indeed out of place, and we removed this sentence in the Conclusion. |
Reviewer 2 Report
Dear dr. Editor
This manuscript is highly relevant in the topics of the Effects of climate and larval habitats on the wing size and 2 shape of male and female avian malaria vector, Culiseta longi- 3 areolata. This study is the first to report its microevolution using geometric morphometry. The primary aim of the study is to determine the intra-specific wing variation between male and female Cs. longiareolata mosquitoes in different climatic and larval habitat types. The manuscript is written with several linguistic and grammar problem from my view;
The materials and methods is well presented as well as the result is well presented and supported with good figures; statistical analysis is support the main ideas and well presnted by figures.
The methodology is complete as:
a. Study Area.
b. Mosquitoes sampling and identification
c. Wing Preparation.
d. Data Analysis
- The conclusion is presented well as" this study showed different types of phenotypic variations between the populations of Cs. longiareolata located in sub-humid and sub-arid regions with a difference in climatic factors: sexual dimorphism and phenotype variations (shape and size of wings) according to larval habitat variability. All control actions in the urban area may play a major role in the fluctuating asymmetry in mosquitoes. The correlations between geographic, climatic, genetic, and phenotypic differentiations should be explored in specific and multidisciplinary works based on larger sample sizes".
- The authors represent his work good and follow the instruction of the journal. but the wing that the authors are drawing must be photographed or supported by figures of the permanent specimens.
The study needs major revision.
Author Response
|
Code |
Comment / Suggestion |
Action taken |
|
R2-01 |
The authors represent his work good and follow the instruction of the journal. but the wing that the authors are drawing must be photographed or supported by figures of the permanent specimens. |
We appreciate the comments by the reviewer. The wing presented in Supplemental Fig S1 is a photograph from an actual specimen. We would like to note that the wing was processed to remove the scales for us to see the wing venation more clearly.
|
|
R2-02 |
The manuscript is written with several linguistic and grammar problems from my view; |
We had the manuscript proofread by an English native at our University. |
Reviewer 3 Report
Manuscript insects-1943296 : Effects of climate and larval habitats on the wing size and 2 shape of male and female avian malaria vector, Culiseta longiareolata
The present study evaluated the effects of of climate and larval habitats on the wing size and shape of male and female avian malaria vector, Culiseta longiareolata.
The paper reports about wing morphological variations in Culiseta longiareolata mosquitoes with respect to different climate regions and larval habitat types using landmark-based geometric morphometrics. Are wing size and shape affected by climate and type of larval habitats? The overall question and the general approach are very good; however, I have reservations about the strength of the evidence included in these observations with respect to the focal question (climate and larval habitat). Specifically, the sample size (e.g. only 91 male and female mosquitoes for both type of habitats for the overall collection period) seems a bit low to draw conclusions about the impact of climate. Moreover, the two collection selected areas are closer to each other (at temperature 10.33°C to 26.1°C vs 12°C to 28°C) and I don’t think these climatic conditions are so different for these observations. Moreover, the Methods part could be improved to describe well the procedures and allow replications by others.
In addition, I include a few additional points below, but I believe that these issues are worthy of consideration before publication.
- I would suggest the title to be rephrased such “Geometric morphometric of wing in Culiseta longiareolata from two locations in Algeria”
- L42: Provide some examples of natural habitats.
- L42- L44: “It was …..Europe and Asia” I suggest to rephrase as: “It was found in all types of natural habitats (e.g…) throughout the country [2]. It is widely distributed in the south of the Palearctic and Mediterranean regions, as well as in Europe and in Asia [3,4].
- In general the method section need to be improved. Eg: With the dipping method, you cannot collect only L4. Give the reference for the dipping method (Silver JB. Mosquito ecology: field sampling methods. New York: Springer; 2008?), clarify whether mosquitoes collected were reared without adding larval food. If yes, this can have a real impact on the size…..
- “Collection of mosquito samples was conducted between January (2020) and July (2021) in thirteen areas 80 distributed in three cities”. It is not clear whether collections in different areas were carried out simultaneously or not and how many collections were carried out. Suggest also to collections were conducted…
- Suggest to keep Table S1 and Fig S1 in the main text of the manuscript. Information in these table and figure are important.
- Provide the distances (in km) between the collection sites
- Provide Relative humidity data of the collection sites
- L92-93: Give also the range of temperature, RH in Annaba
- L120: suggest: “…described by Arnqvist [32]
Author Response
|
Code |
Comment / Suggestion |
Action taken |
|
R3-01 |
I would suggest the title to be rephrased such “Geometric morphometric of wing in Culisetalongiareolata from two locations in Algeria” |
We value the comment of the reviewer. Therefore the new title of the manuscript is “Geometric Morphometric Wing Analysis of avian malaria vector, Culiseta longiareolata from Algeria and its effects on cli-mate and larval habitats” |
|
R3-02 |
L42: Provide some examples of natural habitats. |
modified |
|
R3-03 |
L42- L44: “It was …..Europe and Asia” I suggest to rephrase as: “It was found in all types of natural habitats (e.g…) throughout the country [2]. It is widely distributed in the south of the Palearctic and Mediterranean regions, as well as in Europe and in Asia [3,4]. |
modified |
|
R3-04 |
In general the method section need to be improved. Eg: With the dipping method, you cannot collect only L4. Give the reference for the dipping method (Silver JB. Mosquito ecology: field sampling methods. New York: Springer; 2008?), clarify whether mosquitoes collected were reared without adding larval food. If yes, this can have a real impact on the size….. |
Thank you for this. The authors have modified the section accordingly. Please see lines 107 to 117.
|
|
R3-05 |
“Collection of mosquito samples was conducted between January (2020) and July (2021) in thirteen areas 80 distributed in three cities”. It is not clear whether collections in different areas were carried out simultaneously or not and how many collections were carried out. Suggest also to collections were conducted… |
The authors have included the suggestion in Section 2.2 Please see lines 107 to 108. |
|
R3-06 |
Suggest to keep Table S1 and Fig S1 in the main text of the manuscript. Information in these table and figure are important. |
Thank you for this suggestion however, we would like to still put Table S1 and Fig S1 in the supplemental files. |
|
R3-07 |
Provide the distances (in km) between the collection sites |
The authors have included the suggestion in Section 2.2. Please see lines 111 to 113.
|
|
R3-08 |
Provide Relative humidity data of the collection sites |
The authors have included the suggestion in the methodology section accordingly. Please see lines 91 to 102.
|
|
R3-09 |
L92-93: Give also the range of temperature, RH in Annaba |
The authors have included the suggestion in the methodology section accordingly. Please see lines 94 to 97.
|
|
R3-10 |
L120: suggest: “…described by Arnqvist [32] |
modified |
Round 2
Reviewer 2 Report
This manuscript is highly improved and I recommended publishing it in its form.
Author Response
The authors would like to thank the reviewer for his valuable insights into improving the manuscript.
Reviewer 3 Report
Now I have read your revised manuscript. The revised version was not accompanied with a sentence detailing how the changes were made during revision, therefore it was not clear what changes were made (I tried to understand that. Upon comparing the original and revised text with the reviewers’ comments I have found that the most important comments I made about the sample size and the strenght of the evidence included in these observations about the focal question (climate change and larval habitat) were actually not addressed. Authors should at least respond and discuss these limitations in the discussion section.
Moreover, I am not fully agree with the suggested title 'Geometric morphometric wing analysis of avian malaria vector, Culiseta longiareolata from Algeria and its effects on climate and larval habitats'. How does geometric morphometric analysis has an effet on climate and larval habitat? I don't think it make sens.
However, I agree that the manuscript contains valuable information.
Thanks.
Author Response
First, the authors would like to apologize for not being clear with the point-by-point changes made in the manuscript. Below is the table coded (e.g. R3-01) accordingly in order to reflect the changes made in the new manuscript. Each code represents the changes made in the revised manuscript.
|
Code |
Comment / Suggestion by Reviewer |
Action taken |
|
R3-01 |
I would suggest the title to be rephrased such “Geometric morphometric of wing in Culisetalongiareolata from two locations in Algeria” |
We value the comment of the reviewer. Therefore, the new title of the manuscript is Geometric morphometric of wing in Culiseta longiareolata from two locations in Algeria |
|
R3-02 |
L42: Provide some examples of natural habitats. |
Modified. Included e.g. lagoon depressions, valleys, ditches. Please see Line 41 |
|
R3-03 |
L42- L44: “It was …..Europe and Asia” I suggest to rephrase as: “It was found in all types of natural habitats (e.g…) throughout the country [2]. It is widely distributed in the south of the Palearctic and Mediterranean regions, as well as in Europe and in Asia [3,4]. |
Modified. Please see Lines 41-43 |
|
R3-04 |
In general the method section need to be improved. Eg: With the dipping method, you cannot collect only L4. Give the reference for the dipping method (Silver JB. Mosquito ecology: field sampling methods. New York: Springer; 2008?), clarify whether mosquitoes collected were reared without adding larval food. If yes, this can have a real impact on the size….. |
Thank you for this. The authors have modified the section accordingly. Please see lines 107 to 117.
|
|
R3-05 |
“Collection of mosquito samples was conducted between January (2020) and July (2021) in thirteen areas 80 distributed in three cities”. It is not clear whether collections in different areas were carried out simultaneously or not and how many collections were carried out. Suggest also to collections were conducted… |
The authors have included the suggestion in Section 2.2Please see lines 107 to 108. |
|
R3-06 |
Suggest to keep Table S1 and Fig S1 in the main text of the manuscript. Information in these table and figure are important. |
Thank you for this suggestion however, the authors decided to still put Table S1 and Fig S1 in the supplemental files. |
|
R3-07 |
Provide the distances (in km) between the collection sites |
The authors have included the suggestion in Section 2.2. Please see lines 109 to 111.
|
|
R3-08 |
Provide Relative humidity data of the collection sites |
The authors have included the suggestion in the methodology section accordingly. Please see lines 91 to 102.
|
|
R3-09 |
L92-93: Give also the range of temperature, RH in Annaba |
The authors have included the suggestion in the methodology section accordingly. Please see lines 94 to 97.
|
|
R3-10 |
L120: suggest: “…described by Arnqvist [32] |
modified |
|
R3-11 |
Now I have read your revised manuscript. The revised version was not accompanied with a sentence detailing how the changes were made during revision, therefore it was not clear what changes were made (I tried to understand that. Upon comparing the original and revised text with the reviewers’ comments I have found that the most important comments I made about the sample size and the strenght of the evidence included in these observations about the focal question (climate change and larval habitat) were actually not addressed. Authors should at least respond and discuss these limitations in the discussion section. |
Please see the revised changes in the manuscript as coded.
We also added a section on limitations in the discussion section which addresses the uneven sample sizes.
Please see Lines 335 to 349
|
|
R3-12 |
Moreover, I am not fully agree with the suggested title 'Geometric morphometric wing analysis of avian malaria vector, Culiseta longiareolata from Algeria and its effects on climate and larval habitats'. How does geometric morphometric analysis has an effet on climate and larval habitat? I don't think it make sens. |
Please see R3-01 |